# From Color-Avoiding to Color-Favored Percolation in Diluted Lattices

**Michele Giusfredi** [1,†], **Franco Bagnoli**[1,2,†,*]

1   Department of Physics and Astronomy and CSDC, University of Florence, via G. Sansone 1,
    50019 Sesto Fiorentino, Italy; michele.giusfredi@stud.unifi.it
2   Istituto Nazionale di Fisica Nucleare, sez. Firenze, Via G. Sansone 1, 50019 Sesto Fiorentino (FI), Italy
*   Correspondence: franco.bagnoli@unifi.it
†   All authors contributed equally to this work.

**Abstract:** We study the problem of color-avoiding and color-favored percolation in a network, i.e., the problem of finding a path that avoids a certain number of colors, associated with vulnerabilities of nodes or links, or is attracted by them. We investigate here regular (mainly directed) lattices with a fractions of links removed (hence the term "diluted"). We show that this problem can be formulated as a self-organized critical problem, in which the asymptotic phase space can be obtained in one simulation. The method is particularly effective for certain "convex" formulations, but can be extended to arbitrary problems using multi-bit coding. We obtain the phase diagram for some problem related to color-avoiding percolation on directed models. We also show that the interference among colors induces a paradoxical effect in which color-favored percolation is permitted where standard percolation for a single color is impossible.

**Keywords:** percolation; directed percolation; multi-graphs; color-avoiding percolation

## 1. Introduction

Percolation theory concerns the flow and the diffusion of some quantity on lattices or networks, for instance a disease on a human network or a message in a communication one [1–3]. It has many applications in the Internet science [4], for instance in the problem of robustness under attack [5], or the resilience after a random failure of nodes and/or links [6]. Classical percolation theory studies the penetration of a scalar quantity on a lattice, but it may be extended to more complex situations [7].

Percolation models have been also widely used to model the spreading of a disease [8]. In such models, nodes correspond to individuals that can be in several states. The simplest situation is with two states, susceptible (healthy) and infectious (ill). In this model infectious individuals recover after some time and become susceptible again (susceptible-infectious-susceptible or SIS model). Alternatively, one can introduce a recovered (or dead) state, which does not spread the disease nor can be infected again (susceptible-infectious-recovered or SIR model), many other possibilities exists [9–11]. Links between nodes represent contacts capable of transmitting the disease among individuals: a susceptible individual connected to an infectious one can became infected with a certain probability. The percolation transition represents the outbreak of an epidemic. Similar approaches can be used to model the propagation of computer viruses [12,13].

In many real-world networks, one can identify different classes of nodes that share the same vulnerabilities. For example, one can classify Internet routers that run different software versions in separate classes. Routers that are vulnerable to a certain bug may simultaneously fail and suddenly isolate parts of the network.

In communication networks, vertices can be associated with servers controlled by different companies. Some of them could be interested in eavesdropping information that passes through their

servers, which therefore must be avoided for a secure communication. Even if all the companies are dangerous, a secure communication between the source and the receiver can be achieved by splitting messages in pieces by secret-sharing, and sending them through multiple paths, each one avoiding one of the vulnerable classes of nodes.

The color-avoiding percolation (CAP) theory is a recent generalization of the percolation theory, useful to treat these problems [14–17]. Every node of the network is marked with a one or more colors, associated with the possible vulnerabilities. The goal of CAP is to find whether one or more paths exist between couples of nodes avoiding one or more colors.

Each color can be associated with a layer in a multi-graph, so that the problem of color-avoidance is that of deciding is there exists a path that avoids all nodes or links that are "occupied" in some of the layers.

On the other side, one could profit of nodes that act as gateways towards different companies or alternative networks, seen as connections among layers, and so that in case of failure of one layer they could route the traffic to another one. Associating colors to layers, one could consider the problem of color-favored percolation (CFP), i.e., percolation with the possibility of passing from one layer to another in case of co-occupation. In principle, the two problems could be considered to be dual, by reverting the interpretation of occupation/vacancy of a node for a given color.

In this paper, we show how CAP/CFP problems on direct lattices can be mapped into a self-organized critical (SOC) problem. We study the critical properties of CAP in a direct lattice using different SOC methods, in particular with the fragment method [18].

We also show that the interference among colors in color-favored percolation allows the presence of a percolating cluster that "jumps" among layers where the percolation of a single color is impossible.

We shall deal here with regular lattices with dilution, i.e., with a fraction of links removed. We shall show that the results of simulations on such diluted lattices are very similar to that obtained on random networks, with the same average connectivity.

## 2. Percolation and Directed Percolation

Let us recall some definitions. We denote by $s_i(t) \in \{0,1\}$ the state of site in position $i$ ($1 \le i \le N$) at time $t$ ($0 \le t \le T$), where one means wet or infected and zero dry or healthy. The index $i$ identifies a site in a given network, connected to other sites $j$ as specified by the adjacency matrix $a$: $a_{ij} = 1$ if there is a link from site $j$ to site $i$ and zero otherwise. We denote by $k_i = \sum_j a_{ij}$ the (input) connectivity of site $i$. Let us suppose for the moment that the lattice has constant connectivity $k$, and take the simplest case $k = 2$, with each site $s_i$ connected to its neighboring two sites $s_{i-1}$ and $s_{i+1}$.

In classical percolation, the evolution of the state of the whole system is given by the parallel application of the rule

$$s_i(t+1) = [r_i < p] \wedge (s_{i-1}(t) \vee s_{i+1}(t)) \vee s_i(t), \tag{1}$$

where $r_i$ is a random number uniformly distributed in the interval $[0,1)$, $\vee$ is the OR operation, $\wedge$ the AND one (equivalent to the multiplication) and $[\cdot]$ is the truth function which takes the value one if $\cdot$ is true, and zero otherwise.

Equation (1) essentially says that if a site is wet, it stays wet, and that a dry site can become wet with probability $p$ if at least one of neighbors is so ($[r_i < p]$ gives the probability that the "pore" $s_i$ is open).

The connected wet sites form a percolation cluster. When the probability $p$ goes beyond a critical value $p_c$, a giant percolation cluster appears, spanning the systems, while small isolated clusters are still present. This is the percolation transition.

For modeling a disease spreading, one needs a modification, since the infection is a random event that can occur at every time step, so we modify Equation (1) as

$$s_i(t+1) = [r_i(t) < p] \wedge (s_{i-1}(t) \vee s_{i+1}(t)) \vee s_i(t), \tag{2}$$

where now $r_i(t)$ is independently extracted for each site and each time step. In the language of disease spreading, it is an SIR (Susceptible-Infected-Recovered) model. Since in this and the following cases the "properties" of the lattice (the random numbers) are changing in time, they are also denoted "directed" percolation models [19] (time is the "directed" direction). A given set of random numbers $r_i(t)$ is also called a "realization" of the disorder in the lattice.

Finally, as happens in some disease and in many computer cases, the "ineffectiveness" or misbehaving can be only transient, so after some time a node returns to the initial state, i.e., a SIS (Susceptible-Infected-Susceptible) model. This has the following implementation,

$$s_i(t) = [r_i(t) < p] \wedge \left(s_{i-1}(t-1) \vee s_{i+1}(t-1)\right). \tag{3}$$

We shall deal here essentially with similar SIS models. They can be generalized, for instance, one can have an infection probability that increases with the number of infected neighbors, or even cases in which there is an interference effect among them, such as in the Domany–Kinzel (DK) model [20,21],

$$s_i(t) = \left[r_i(t) < \tau\left(s_{i-1}(t-1) + s_{i+1}(t-1)\right)\right] \tag{4}$$

where now the probabilities $\tau(n)$, $n = 0, 1, 2$ (number of ones in the neighborhood) define the model. In the example of Equation (3), $\tau(0) = 0$; $\tau(1) = \tau(2) = p$.

In general, one assumes $\tau(0) = 0$, so that the configuration $s_i = 0 \; \forall i$ becomes an absorbing state [22]. In the DK model, one defines $\tau(1) = p$ and $\tau(2) = q$ (Figure 1). The order parameter is the asymptotic "density" $\rho = \lim_{t\to\infty} \rho(t)$ of active (non-zero) ($s_i(t) = 1$) sites, where

$$\rho(t) = \frac{1}{N} \sum_i s_i(t),$$

which is zero below $p_c$ and greater than zero above.

Let us introduce some useful terms. We refer to the progress of the percolation process with the term "infection" and to the region of phase space where in the long-time limit there is a finite probability of survival for the percolation process as the "active" region ($\rho > 0$), see Figure 1.

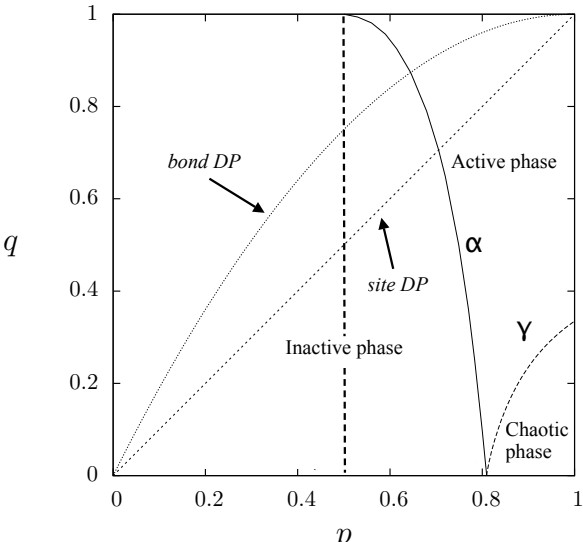

**Figure 1.** The phase diagram of the DK model, $\alpha$ marks the critical line separating the active and inactive (or absorbing) phase, $\gamma$ marks the "chaotic" phase near the corner $p = 1, q = 0$. The simplest mean-field approximation gives $p_c = 1/2$ independently of $q$.

One can show that the standard site (directed) percolation corresponds to the line $q = p$ and that the bond (directed) percolation corresponds to the curve $q = p(1 - 2p)$, but other interesting behaviors occurs in the $p - q$ phase space, in particular near the corner $p = 1$, $q = 0$ (chaotic phase, Figure 1) that corresponds to a disruptive interference for which percolation can happen more easily if there is just one "infected" neighbors than if there are two [23,24].

Other extensions are possible, for instance the infection process may depend on the perception of the risk of being infected [25], which has application to computer networks [26].

## 3. Self-organized Criticality and the Fragment Method

In systems showing a continuous phase transition, like the percolation ones, order parameters such as the asymptotic density of active sites $\rho(p)$ show a power-law behavior in the vicinity of the critical value $p_c$:

$$\rho(p) \propto (p - p_c)^\beta,$$

which is a mark of the self-similar (scale-free) character of the transition.

While in "standard" critical systems this power-law behavior can be observed only via a fine triggering of parameters, the so-called auto-critical systems self-organizes into a critical state, which is therefore denoted as "self-organized criticality", to underline their ability to "go freely" to the critical point.

Known examples of auto-critical systems have been used for modeling earthquakes [27] and evolution [28], and the invasion percolation model [29].

Some classic percolation problems can be reformulated as SOC system [30], and the evolution of certain quantities can be used to obtain information on the criticality of the systems, such as the critical values of their control parameters.

We shall illustrate this idea for the simple directed site percolation problem with $k = 2$. Let us introduce the quantity $p_i(t)$, which is the minimum value of $p$ for which the giant percolation cluster includes site $i$ at time $t$. The quantity $s_i(t)$ can therefore be written as $[p > p_i(t)]$, and Equation (3) as

$$[p > p_i(t)] = [p > r_i(t)] \wedge ([p > p_{i-1}(t-1)] \vee [p > p_{i+1}(t-1)]), \tag{5}$$

The expression $[p > a] \vee [p > b]$ can be rewritten as $[p > \min(a,b)]$ and $[p > a] \wedge [p > b]$ is equivalent to $[p > \max(a,b)]$. Equation (5) becomes

$$[p > p_i(t)] = \left[ p > \max\left(r_i(t), \min(p_{i-1}(t-1), p_{i+1}(t-1))\right) \right].$$

We can therefore extract the evolution rule for the $p_i(t)$:

$$p_i(t) = \max\left(r_i(t), \min(p_{i-1}(t-1), p_{i+1}(t-1))\right),$$

which has a sort of magic: automatically (or self-critically) the $p_i$'s evolve so that all of them are above the critical value $p_c$, for the given lattice and the set of random number used, since, by definition, the critical percolation probability is the smallest value of $p$ for which there is a giant percolation cluster, spanning the whole system.

Thus, $p_c$ can be approximated by

$$p_c = \underset{0 < i \leq N}{\mathrm{MIN}} \, p_i(t),$$

This estimation is just an approximation, since one should consider the thermodynamic ($\lim_{t \to \infty} \lim_{N \to \infty}$) limit. Moreover, the previous result depends on the realization of the disorder, i.e., the set of random numbers used for the simulation, and one should average over different realizations. In practice, the variance of the critical values of $p_c$ obtained in different realizations

always drops to zero when increasing the system size, showing the presence of strong self-averaging in these systems [31].

This method is powerful and elegant, but it can be used only if the rule can be expressed using the AND and OR operation.

It is possible to think to sites as segments initially marking all values of $p$, from zero to one. Since site $i$ can be wet only for $p > p_i$, the associated segment can be painted with white (dry) from zero to $p_i$, and black (wet) above it. At beginning, all segments are black, meaning that initially all are wet.

The AND operation is similar to taking the intersection of the black parts of the segments, and the OR to the union of them. Since these operations keep the black part contiguous, one has only to keep track of the lower boundary, which is exactly $p_i(t)$ [18].

Clearly, if the rule includes also other operations like NOT or XOR, the black part of the segment is no more compact.

We can however approximate the evolution of a system by iterating segments (now called fragments) sampled at many values of $p$, using a multi-bit technique. In this way we can compute the parallel evolution of the system for many values of $p$ over the random field determined by the random numbers $r_i(t)$, as shown in Ref. [18] (the fragment method).

This method can be applied to any system that can be described as a probabilistic cellular automaton (PCA). In a network we can describe the evolution of the states $s_i(t)$ of the nodes with the local evolution rule

$$s_i(t+1) = f(\{s_i(t)\}; \{[r_i(t) < p]\}),\qquad(6)$$

in which time $t$ is a discrete variable, and $f$ is a function of the set of states $\{s_i(t)\}$ of the nodes connected to node $i$ and of the set $\{[r_i(t) < p]\}$ of random numbers confronted with the control parameters.

The important aspect of this procedure is that the evolution of the fragments is independent of the values of the control parameter $p$. The parameter is reintroduced in the problem only in the final step of the procedure, to pass from the fragment configuration to the phase-space configuration.

The evolution of the fragments can be seen as the evolution in parallel of different copies of the original system over the same disordered field (the random numbers). The different values of the control parameter correspond to copies of the system sharing the same initial configuration and the same set of random numbers. The bits of the multi-bit representation of fragments are the values of the variables $s$ for each of the values of the control parameter. The percolation cluster for those systems with control parameter less than the critical value $p_c$ is finite (does not reach a large time), so that what happens is that fragments self-organize so that (for large enough systems) they are zero for $p < p_c$ and different from zero above. By inspecting the fragments, one can get an approximate value of $p_c$.

One can also consider a probabilistic system characterized by $m$ control parameters $p_j$, with $j = 1, 2, \ldots, m$. In this case, the fragment method consists of assigning a fragment in $m$ dimensions to every node. The fragments in $m$ dimensions are defined as subsets of $m$-dimensional unit hypercube. A fragment in 2 dimensions, for example, is a subset of the unit square $[0,1) \times [0,1)$.

## 4. Color-Avoiding Percolation

The idea of color-avoiding percolation is that of assigning one or more colors (exclusive or co-present) to sites and/or bonds, and searching for a path that avoids them.

*Site and Bond Color-Avoiding Percolation*

Let us assign a color $c_i \in \{1, 2, \ldots, C\}$ to each vertex $i$ of a network. The quantity $C$ denotes the total number of colors. Assuming that the sets of nodes with a certain color are disjoint, we can define $q_c$ the probabilities that a node has the color $c$, such that $\sum_c q_c = 1$.

There are two possibilities:

- Independent colors: nodes can have at the same time more than one vulnerability/color;

- Exclusive colors: every node can assume only one color (cases studied in Refs. [14–16]).

We assume that nodes with the same color fail together and that those with different colors unlikely fail at the same time. Some colors can be considered trusted, and it is not necessary to avoid them for sending, receiving or transmitting some information.

We then define a pair of nodes as color-avoiding connected (CAC) if there are paths between them avoiding nodes carrying untrusted colors, with each path avoiding a different color (the same path can be used to avoid more than one color).

We can also define the color-avoiding giant cluster (CAGC) as the maximal set of nodes that are mutually CAC, i.e., the largest set of nodes connected to each other and such that any pair of vertices belonging to the set is CAC. A method to extract this set from the network can be found in Ref. [14].

As with site color-avoiding percolation, two nodes can be defined as pair CAC if they are connected through paths that avoid edges of a certain color, thus defining a bond CAP problem.

## 5. Applications

Simulations have been made on lattices with $N$ sites in their directed formulation, i.e., with random variables renewed at each time step. A site $i$ is connected to $k$ other sites.

Basic results obtained by means of the SOC method are reported in Ref. [17].

Let us report first the most important results on bond CAP with exclusive colors. We consider a network in which edges are colored with one of two colors, for example red (r) and blue (b). We also considered a dilution of the network with the parameter $\phi$, the probability that an edge is not removed from the network. The local evolution rule can be written as

$$\begin{cases} s_i^{(\bar{r})}(t+1) = \bigvee_{j:a_{ij}=1} [\phi > \eta_{ij}(t)][p < r_{ij}(t)]s_j^{(\bar{r})}(t) \\ s_i^{(\bar{b})}(t+1) = \bigvee_{j:a_{ij}=1} [\phi > \eta_{ij}(t)][p > r_{ij}(t)]s_j^{(\bar{b})}(t) \end{cases} \tag{7}$$

where the states $s_i^{(\bar{r})}$ and $s_i^{(\bar{b})}$ are one if the site $i$ is not red (resp. not blue). They evolve at the same time with the same random variables $\eta_{ij}(t)$ and $r_{ij}(t)$, associated with the edges connecting node $j$ to node $i$. The dilution parameter $\phi$ is confronted with $\eta_{ij}(t)$ in the first truth function of the two equations to determine whether the connection is removed or not, while $p$ is the probability that a node is red (otherwise it is blue), the control parameter to be compared with $r_{ij}(t)$.

In this case, the product (AND) between the two states $s_i^{(\bar{r})}$ and $s_i^{(\bar{b})}$ is of particular interest. Indeed, the product can be equal to 1 only if the node $i$ is reached through at least two paths, one that avoids the red edges, and one avoiding the blue ones. Since each link carries one of the two colors, the two paths have no links in common.

Imposing as initial condition that the states $s_i^{(\bar{r})}(0)$ and $s_i^{(\bar{b})}(0)$ are equal to 1 for all the nodes, after a sufficient amount of time the product will be non-zero only for a fraction of the nodes of the network that can be interpreted as elements of the CAGC (Figure 2).

We have studied the value that assumes this product $\mathcal{M} = \frac{1}{N}\sum_i s_i^{(\bar{r})}(t)s_i^{(\bar{b})}(t)$ in the parameter space, as shown in Figure 2. The phase planes were in this case obtained with a single simulation of the evolution of the fragments in a lattice with $N = 10,000$, at the asymptotic time $t = 10,000$, and varying the connectivity $k$ of the nodes.

If the connectivity $k$ is lower than 3, it is not possible to find nodes such that $s_i^{(\bar{r})}(t)s_i^{(\bar{b})}(t) = 1$ at large times, regardless of the values of the control parameters. Instead, for $k \geq 3$ there are certain values of $\phi$ and $p^{(1)}$ for which $\mathcal{M}$ undergoes a phase transition and passes from 0 to a non-null-value. Increasing the value of $k$, the region in the graph in which $\mathcal{M} \neq 0$ increases its size, and for example with $k = 20$, as shown in Figure 2, it includes the majority of the phase plane.

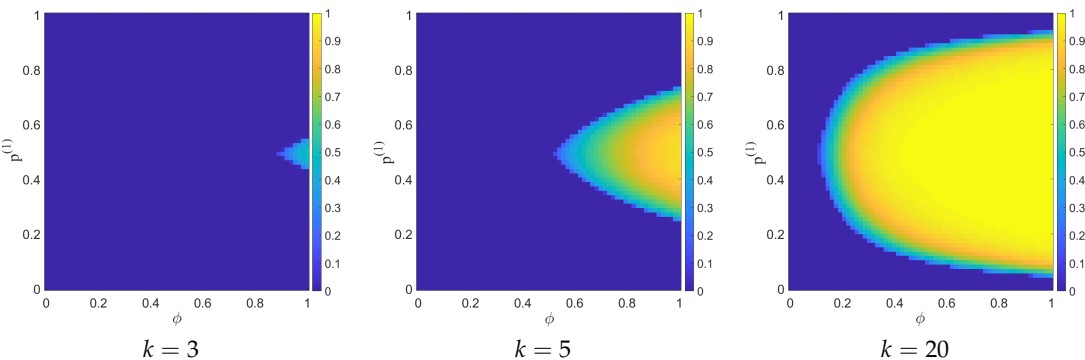

**Figure 2.** Phase planes of the bond CAP for two exclusive colors (denoted color 1 and 2). assigned to sites with probability $p(1)$ and $p(2) = 1 - p(1)$ and different dilutions (different connectivities).

These figures are very similar to those reported in Ref. [15], obtained using an Erdős–Rényi network, showing that indeed in these cases the relevant parameter is the average connectivity.

The Equation (7) can be generalized to describe the evolution of networks with a generic number of exclusive colors $C$. In this case, it is useful to define the random variables

$$\mathcal{C}_c(r) = \left[ \sum_{j=0}^{c} p(j) \leq r < \sum_{j=0}^{c+1} p(j) \right] = \left[ r \leq \sum_{j=0}^{c} p(j) \right] \oplus \left[ r < \sum_{j=0}^{c+1} p(j) \right], \quad c \in \{1, 2, \cdots, C\}, \quad (8)$$

where $p(c)$ is the probability of having color $c$ ($p(0) = 0$) and $\sum_{c=1}^{C} p(c) = 1$.

Considering a random variable $r$, associated with and edge connecting two nodes, and a color $c$, the variable $\mathcal{C}_c(r)$ is equal to 1 with probability $p(c)$ and equal to 0 with probability $1 - p(c)$, thus allowing us to determine if the edge is colored with color $i$, or another color.

In addition, if $\mathcal{C}_c(r) = 1$ for a certain value of $r$, i.e., the edge has the color $c$, then $\mathcal{C}_j(r) = 0$ for every other color $j \neq c$, hence the edge cannot have two different colors at the same time.

Therefore, for a generic network with $C$ exclusive colors, and some color $c$ to be avoided, the bond color-avoiding percolation can be studied with the following local evolution rule

$$s_i^{(\bar{c})}(t+1) = \bigvee_{j:a_{ij}=1} [\phi > \eta_{ij}(t)] \overline{\mathcal{C}_c}(r_{ij}(t)) s_j^{(\bar{c})}(t+1) \quad \forall c, \quad (9)$$

where

$$\overline{\mathcal{C}_c}(r) = 1 \oplus \mathcal{C}_c(r) \quad (10)$$

is the negation of $\mathcal{C}_c(r)$.

As for the case of the bond CAP with two exclusive colors, the random variable $\eta_{ij}(t)$ is confronted with the dilution parameter $\phi$ to determine whether the connection between node $i$ and $j$ is removed or not, while $r_{ij}(t)$ determines the color of the edge. We can consider again the product of the states $s_i^{(\bar{c})}$ associated with all the colors $c$ to be avoided, which is equal to one only if node $i$ can be reached avoiding every untrusted color, and the fraction of nodes for which this product is equal to one, denoted as

$$\mathcal{M}(t) = \frac{1}{N} \sum_i \left( \prod_c s_i^{(\bar{c})}(t) \right). \quad (11)$$

Imposing as initial condition $s_i^{(\bar{c})}(0) = 1$ for all nodes and all colors, the asymptotic value $\lim_{t \to \infty} \mathcal{M}(t)$ can be interpreted as the relative size of the CAGC.

Using Equation (9), we studied the bond CAP in the case of three exclusive colors, and obtained the phase planes shown in Figures 3 and 4. The phase planes were obtained with a single simulation of the evolution of the fragments in a lattice with $N = 10,000$, at the asymptotic time $t = 100,000$.

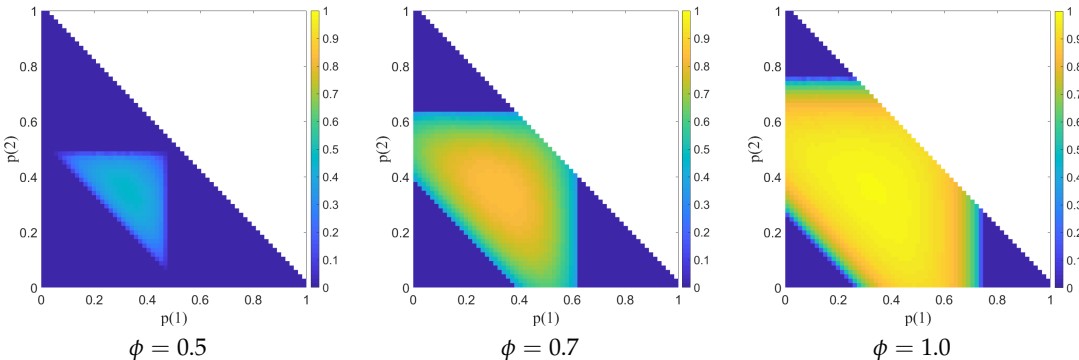

$\phi = 0.5$ $\qquad\qquad$ $\phi = 0.7$ $\qquad\qquad$ $\phi = 1.0$

**Figure 3.** Phase planes of the bond CAP for three exclusive colors, in networks with average connectivity $k = 5$, at different values of the dilution parameter $\phi$.

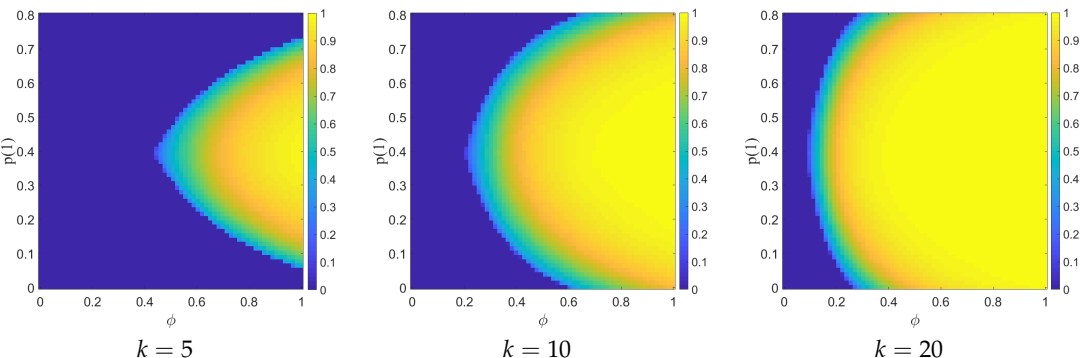

$k = 5$ $\qquad\qquad$ $k = 10$ $\qquad\qquad$ $k = 20$

**Figure 4.** Phase planes of the bond CAP for three exclusive colors, with one trusted color with fixed probability $p(3) = 0.2$, varying the average connectivity $k$ of the network.

For the phase planes shown in Figure 3 we considered the case in which the tree colors $(1 - 3)$ are untrusted, and we studied the values of $\mathcal{M}$ in the parameter space, that can be represented as the set of allowed values of the color probabilities $p(1)$ and $p(2)$ such that $\sum_{c=1}^{3} p(c) = 1$, and $p(c) \geq 0 \quad \forall c$ (the probability for the third color, $p(3)$, is fixed given the values of $p(1)$ and $p(2)$ since $p(3) = 1 - p(1) - p(2)$), in networks with average connectivity $k = 5$, at different values of the dilution parameter $\phi$.

As it can be seen, $\mathcal{M} = 0$ near the three vertices of the "triangle" of the allowed values of $p(1)$ and $p(2)$, that corresponds to the situations in which one of the three colors prevails in the network and is hard to avoid, and the other two are less present, as for example for $p(1) \sim 0$, $p(2) \sim 0$, while $p(3) \sim 1$.

Far from the vertices, the system can be in the active phase with $\mathcal{M} \neq 0$, and its size in the phase plane depends on the connectivity of the network, as it can be seen considering different values of the dilution parameter.

The size of the active phase decreases lowering the value of $\phi$, and for small enough values as $\phi = 0.5$, if one color is nearly absent from the network and easy to avoid, as for example for $p(1) \sim 0$, the other colors become harder to avoid, and $\mathcal{M} = 0$ for all the values of $p(2)$ and $p(3)$.

We also studied a different problem for the phase planes, considering networks with one trusted color and two untrusted ones. In this case, colors 1 and 2 are to be avoided, while it is not necessary to

avoid the third color, therefore $\mathcal{M}$ is dependent on the product of the states associated with only the first two colors, Figure 4.

The phase planes have been obtained by fixing the value of the probability of the third color to $p(3) = 0.2$, considering different values of the average connectivity $k$, and studying the value of $\mathcal{M}$ in the parameter space of the dilution parameter $\phi$ and the allowed values of the color probability $p(1)$ (that cannot be greater than 0.8, because of the presence of the third color, while the probability for the second color is fixed as $p(2) = 1 - p(1) - p(3)$).

This phase planes are clearly similar to those in Figure 2, as the exchange symmetry of the two untrusted colors is preserved by the presence of the third color, and the smaller critical value of $\phi$ is still obtained for $p(1) = p(2)$. However, while in the case of two exclusive colors it is not possible to have $\mathcal{M} \neq 0$ if one of the untrusted colors is nearly absent from the lattice, as the other untrusted one becomes impossible to avoid, even for great values of $k$, in the case of three exclusive colors, the presence of a trusted color can allow $\mathcal{M} \neq 0$ for $p(1) = 0$ or $p(2) = 0$, if the connectivity $k$ is large enough.

## 6. Multilayer Model

We can have an alternative view of color-avoiding percolation by mapping it onto a multilayer directed process.

Let us consider two (or more) layers, defined by an index $c$, $c = 1, \ldots, C$, so that now the equations concerns quantities such as $s_i^{(c)}(t)$ and random numbers $r_i^{(c)}(t)$. We can assume that each layer corresponds to a given color. Let us first discuss the color-favored situation, in which the percolation clusters are composed by sites carrying at least one color. Equation (4) becomes

$$s_i^{(c)}(t) = [r_i^{(c)}(t) < \tau(S_{i-1}(t-1) + S_{i+1}(t-1))], \tag{12}$$

where for color-favored percolation,

$$S_i(t) = \bigvee_c s_i^{(c)}(t).$$

Sites with more than one color act as gateways among layers.

For color-avoiding percolation and not-exclusive colors, Equation (4) becomes

$$s_i^{(c)}(t) = [r_i^{(c)}(t) < \tau(S_{i-1}^{(c)}(t-1) + S_{i+1}^{(c)}(t-1))], \tag{13}$$

where now $S_i^{(c)}(t)$ denotes the presence of color $c$ and the absence of other colors $c' \neq c$ at site $i$ and time $t$,

$$S_i^{(c)}(t) = s_i^{(c)}(t) \bigwedge_{c' \neq c} \overline{s_i^{(c')}(t)},$$

where the overbar denotes the negation. For exclusive colors, $S_i^{(c)}(t)$ is simply $s_i^{(c)}(t)$, since the presence of one color $c$ implies the absence of other colors $c'$ in that site.

In the following, we shall use the terminology of color-favored percolation, i.e., we shall say that the system percolates when there is activity in the phase space.

Up to now we described the situation in which colors do not interact, so that the color-avoiding percolation phase just corresponds to the absorbing ($\rho^{(c)} = 0$) state of the standard DK model of Figure 1.

What happens when colors interfere? To reduce the number of parameters we assume that the model is symmetric among colors. For two colors ($c = 1, 2$) we have now that the transition probability for layer 1 is $\tau(n^{(1)}, n^{(1)})$, (and symmetrically for layer 2 is $\tau(n^{(2)}, n^{(0)})$). We again assume that the empty state is absorbing, so that $\tau(0, 0) = 0$.

If we assume that the presence of a color is sufficient to stop percolation on that layer, we have the same model as before except for the case in which there is no color on one layer but there is one on the other: $\tau(1,0) = \tau(1,1) = \tau(1,2) = p$ and $\tau(2,0) = \tau(2,1) = \tau(2,2) = q$. We are left with two free parameters: $\tau(0,1)$ and $\tau(0,2)$, which express the "interference" for which a color on a layer can affect the percolation on the other layer.

We studied two cases: $\tau(0,1) = \tau(0,2) = \varepsilon$ and $\tau(0,1) = \varepsilon$, $\tau(0,2) = 0$. The first case (a) is similar to site percolation, while the other (b) reminds the "disruptive interference" of the DK model.

For small enough values of $\varepsilon$ nothing new happens, since in the absorbing phase the "activity" on both layers goes to zero, and in the "active" phase, the influence of the interference term $\varepsilon$ is small. The only influence of $\varepsilon > 0$ is that of shifting the border of the active phase (the simplest mean-field analysis predicts that the threshold $p_c = 1/2$ for the DK model now becomes $p_c = 1/2 - \varepsilon$).

However, when $\varepsilon$ is about $0.7 - 0.75$, one can see that the phase boundary becomes more complex: an active phase appears near the corner $p = 0, q = 0$ for case (a), and this extend to the whole side $p = 0$ for case (b).

What happens here is that the "percolation paths" jump from one layer to the other. Indeed, the value $\varepsilon \simeq 0.7$ is near the site percolation threshold for the standard DK model, so that even an isolated infected site on a layer can propagate to the other layer, although is almost-immediately disappearing on the original layer due to the small values of $p$ and $q$. Actually, the survival on the original layer is detrimental for percolation, as shown by the appearance of an active island near the corner $p = q = 0$ in Figures 5 and 6.

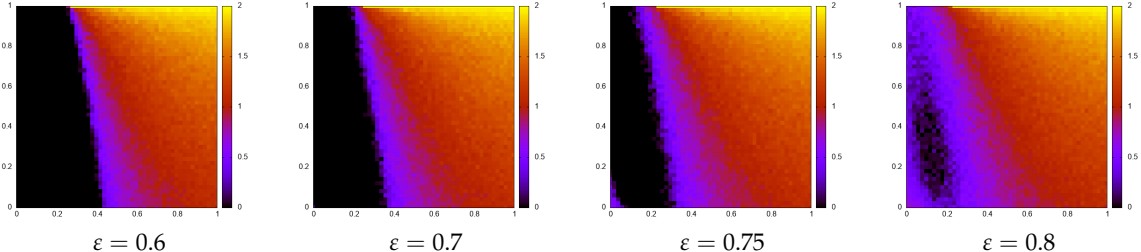

**Figure 5.** Phase space of directed percolation with interacting colors, case (a). The vertical axes is $p$ and the horizontal one is $q$. For moderate values of $\varepsilon$ one simply has a shift of the critical line, but for $\varepsilon \geq 0.75$ a region near the corner $p = q = 0$ becomes active.

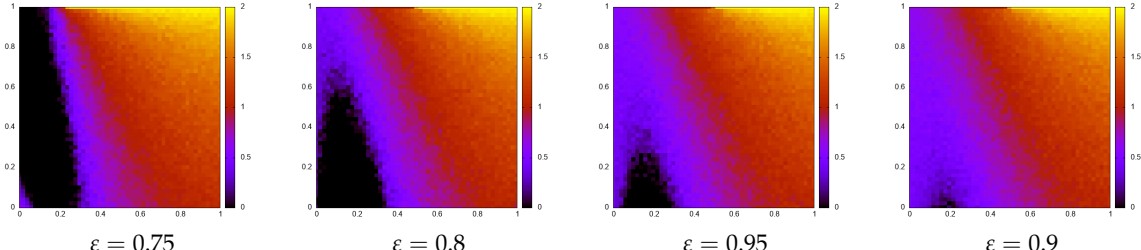

**Figure 6.** Phase space of directed percolation with interacting colors, case (b). The vertical axes is $p$ and the horizontal one is $q$. In this case, (disruptive interference among layers) the active region tends to occupy the whole line $p = 0$.

What happens here is that an isolated active site can percolate on the other layer, but can also survive on the original layer, according to the values of $p$ and $q$. However, as far as it survives, the two layers are decoupled, while if it immediately disappears the layer can be re-infected by the other layer (where also the survival probability depends on $p$ and $q$. Therefore, we obtain the paradoxical result that in the case of enough interfering colors, color-favored percolation is favored where single-color percolation is least surviving. Clearly, the opposite happens for color-avoiding percolation.

## 7. Conclusions

We investigated the problem of color-avoiding and color-favored percolation on a diluted lattice [14–16], i.e., the problem of finding a path that avoids a certain number of colors, associated with vulnerabilities of nodes or links.

We have shown that this problem can be formulated as a self-organized critical problem, in which the asymptotic phase space can be obtained in one simulation, analogous to Invasion percolation [29].

By means of the fragment method [18], we obtained the phase diagram for many problems related to color-avoiding percolation, showing in particular that results obtained for Erdős–Rényi networks can be recovered using the dilution of the rule on regular lattices.

We investigated directed models, and we have shown that the interference among colors in color-favored percolation nor only allows the survival of a percolating cluster that continuously "jumps" among layers where the percolation of a single color is impossible, but even that increasing the survival probability of a cluster of a single color results in lowering the survival probability of the "jumping" cluster.

**Author Contributions:** Conceptualization, F.B. and M.G.; formal analysis, F.B.; methodology, investigation and software, M.G. and F.B.; supervision, F.B.; writing, M.G. and F.B. All authors have read and agreed to the published version of the manuscript.

**Funding:** This research received no external funding.

**Acknowledgments:** The present version is an extension of Ref. [17]. In this version we added more results on color-avoiding percolation, and the part about color-favored percolation.

**Conflicts of Interest:** The authors declare no conflict of interest.

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
