# Peer review of "From Color-Avoiding to Color-Favored Percolation in Diluted Lattices"

_futureinternet, doi:10.3390/fi12080139_

Round 1

Reviewer 1 Report

This paper studied the the  problem of color-avoiding and color-favored percolation in a network, and extended this problem into directed and multilayer models. It contained many interesting models and results, and some theoretically and experimentally results show the effectiveness of those models and results. This paper is interesting. However, the writting and organization  of this paper need to be greatly improved. Moreover, more related works and references should be reviewed. In addition, Please give some correlations between the percolation in this problem and that in the robustness of networks. Finally, more experiments on the other networks such as scale-free networks and small-world networks should be given.

Author Response

> This paper studied the the problem of color-avoiding and color-favored percolation
> in a network, and extended this problem into directed and multilayer models.
> It contained many interesting models and results, and some theoretically
> and experimentally results show the effectiveness of those models and results.
> This paper is interesting. However, the writting and organization of
> this paper need to be greatly improved.

We revised the whole manuscript, changing several expressions

> Moreover, more related works and references should be reviewed.

We added several references

> In addition, Please give some correlations between the percolation in this
> problem and that in the robustness of networks.

Inserted in the Introduction

> Finally, more experiments on the other networks such as scale-free networks
> and small-world networks should be given.

One of the results we obtained is that lattices with dilution show phase diagrams very similar to that obtained on random networks, i.e., a result wuite similar to what is expected with small-world rewiring. We therefore decided to limit the present investigation to lattices  with dilution, now pointed out in the title and the abstract.

Reviewer 2 Report

Too many parts are not presented in an intelligible form. I could not decide if the results are correct and/or of interest.

Here below is a small selection of what is either hard or impossible to read. 

There are 3 main issues:

  1. Many symbols or words are used before their meaning is clarified. 

For example:

Line 56: "infected": no epidemic has been mentioned so far (it is mentioned soon after)

From before Line 65: N is used many times (its meaning is finally indicated in Line 152)

Line 69: "survival" of what?

Figure 2: "p(1)" has not been defined yet (it will be in the next page)

     2. Some expressions and formulas are impossible to understand.

For example:

Formula before Line 65 (this is a central formula in the later and original part of the paper): I believe that s_i(t) takes values either 0 or 1, but \tau is a probability, so in this formula the r.h.s. takes values in the interval [0,1]. I have no idea of what this formula means.

The fragment method is not explained in any detail. Once fragments evolve, what feature of the fragments is looked upon? And then what dimension "m" is going to be used in the later simulations? As no reference is given, Section 3.1 seems a self sufficient description of the method, while clearly it is not. Only in the conclusions (Line 236) does one learn that there is actually a reference where one could gather some more information about this method.

Between Lines 218 and 219: the sentence "in which percolation can occur if at a given site there is at least a color" is incomprehensible. Percolation is global phenomenon, how can it be related to "one given site"? The authors must have meant something else.

Formula (12), which seems one of the main results of the paper, is as incomprehensible as the Formula before Line 65.

Line 118: it is not clear what "it" refers to.

Section 6: there seems to be no "directed" mechanism in the algorithm described in this section, at least I could not detect anything "directed" but Line 281 clearly mentions that something is "directed".

3. Even statements which could have a proof are stated without even a hint to any rigorous justification.

For example:

The statement in the formula before Line 89 (I assume the MIN is taken for 0<i<N+1). Is it proven somewhere?

Author Response

> Too many parts are not presented in an intelligible form. I could not decide if the
> results are correct and/or of interest.

> Here below is a small selection of what is either hard or impossible to read.

> There are 3 main issues:

> Many symbols or words are used before their meaning is clarified.

> For example:

> Line 56: "infected": no epidemic has been mentioned so far (it is mentioned soon after)

Actually, the equivalence between "infected" and "wet" was stated on line 49, but we changed "infected" with "wet" on line 56

> From before Line 65: N is used many times (its meaning is finally indicated in Line 152)

we added the definition of N on line 48

> Line 69: "survival" of what?

We replaced "survival" with "presence"

> Figure 2: "p(1)" has not been defined yet (it will be in the next page)

Figureas are floating objects, they could appear anywhere in the final version,
we added the definition of p(1) in the caption.

> 2. Some expressions and formulas are impossible to understand.

> For example:

> Formula before Line 65 (this is a central formula in the later and original
> part of the paper): I believe that s_i(t) takes values either 0 or 1, but
> \tau is a probability, so in this formula the r.h.s. takes values
> in the interval [0,1]. I have no idea of what this formula means.

We are sorry, the "p" should be replaced by "tau", now corrected.

> The fragment method is not explained in any detail. Once fragments evolve,
> what feature of the fragments is looked upon? And then what dimension "m"
> is going to be used in the later simulations? As no reference is given,
> Section 3.1 seems a self sufficient description of the method, while
> clearly it is not. Only in the conclusions (Line 236) does one
> learn that there is actually a reference where one could gather
> some more information about this method.

Actually, we inserted the reference of line 236 also just before Section 3.1,
but we probably did not add enough emphasis, so we inserted

> Between Lines 218 and 219: the sentence "in which percolation can occur
> if at a given site there is at least a color" is incomprehensible.
> Percolation is global phenomenon, how can it be related to "one given site"?
> The authors must have meant something else.

Yes, indded we used an unclear expression, now changed.

> Formula (12), which seems one of the main results of the paper,
> is as incomprehensible as the Formula before Line 65.

Sorry, same error ad before (corrected)

> Line 118: it is not clear what "it" refers to.

We deeply changed the phrase.

> Section 6: there seems to be no "directed" mechanism in the algorithm described in this section,
> at least I could not detect anything "directed" but Line 281 clearly
> mentions that something is "directed".

We defined the "directedness" in Section 2 (Percolation and Directed Percolation)

> 3. Even statements which could have a proof are stated without even a hint to
> any rigorous justification.

> For example:

> The statement in the formula before Line 89 (I assume the MIN is
> taken for 0<i<N+1). Is it proven somewhere?

We replaced "extracted" by "approximated" and added the definition of the
critical percolation probability

Round 2

Reviewer 1 Report

Many thanks for authors' responses. I have no other comments.

Reviewer 2 Report

The presentation is much more readable in this version.

I still have a few questions (and one main one), listed below together with some typos.

Line 71: If I understand correctly, the model presented in (1) with k=2 (and Dimension 1, as it seems implicitly stated) has a phase transition at p_c=0: nodes i such that r_i>p never become wet, and everything else becomes wet; this is true for all p. This is not relevant in what follows.

Lines 74-78: also in the model (2) the critical point is trivial, as everything becomes eventually wet. This too is not relevant in what follows.

Line 121: "more" --> "longer"

Lines 178-180: It is not entirely clear to me why the nodes in which the product is nonzero should be elements of the CAGC: I understand it is like two  copies, one for each color, and that if there is giant component in each they tend to overlap; but what if these giant components are "small" and overlap very little or nothing?

Line 184 (MAIN ISSUE): It is not entirely clear to me what graph is being used in the simulations. Each node is connected to k nodes, but are they arranged in a one dimensional array? And is any node connected to the adjacent ones? This was the case in Section 3. Is the geometrical arrangement of any relevance? It does not seem to be, as you mention that you have results similar to those for the Erdos-Renyi complete graph.

Line 193: "and" --> "an"

Lines 204-205: same as the main issue above.

Line 290: "nor" --> "not"